# JAK2 Phosphorylation Signals and Their Associated Cytokines Involved in Chronic Rhinosinusitis with Nasal Polyps and Correlated with Disease Severity

**DOI:** 10.3390/biom11071059

**Published:** 2021-07-20

**Authors:** Yu-Tsai Lin, Wei-Chih Chen, Ming-Hsien Tsai, Jing-Ying Chen, Chih-Yen Chien, Shun-Chen Huang

**Affiliations:** 1Department of Otolaryngology, Kaohsiung Chang Gung Memorial Hospital and Chang Gung University, College of Medicine, Kaohsiung 833, Taiwan; xeye@cgmh.org.tw (Y.-T.L.); jarva@cgmh.org.tw (W.-C.C.); b9302094@cgmh.org.tw (M.-H.T.); 2Kaohsiung Chang Gung Head and Neck Oncology Group, Cancer Center, Kaohsiung Chang Gung Memorial Hospital, Kaohsiung 833, Taiwan; 3College of Pharmacy and Health Care, Tajen University, Pingtung County 907, Taiwan; 4Department of Pathology and Lab Medicine, Kaohsiung Veterans General Hospital, Kaohsiung 813, Taiwan; jy@vghks.gov.tw; 5Department of Anatomic Pathology, Kaohsiung Chang Gung Memorial Hospital and Chang Gung University, Kaohsiung 833, Taiwan

**Keywords:** chronic rhinosinusitis, nasal polyps, p-JAK2, p-STAT, disease severity

## Abstract

Janus kinase 2 (JAK2) is a member of the JAK family that transduces cytokine-mediated signals via the JAKs/STATs (signal transducer and activator of transcription proteins) pathway, which plays an important role in many inflammatory diseases. This study investigates the association of p-JAK2 and JAK2-associated cytokines from nasal polyp (NP) tissue with disease severity, and evaluates the p-JAK2-mediated STATs in chronic rhinosinusitis (CRS) with NP. Sixty-one CRSwNP patients with nasal polyps undergoing endoscopic sinus surgery were enrolled, while the turbinate tissues from 26 nasal obstruction patients were examined as the control group. Elevated levels of p-JAK2 were detected in CRSwNP, and significantly correlated with scores of disease severity (LMK-CT, TPS, and SNOT-22). Expressions of the JAK2-associated cytokines, such as IL-5, IL-6, IL-13, G-CSF, and IFN-γ were significantly higher in CRSwNP than in the controls, while the levels of IL-5, IL-6, IL-13, or G-CSF had positive correlation with scores of disease severity. Moreover, markedly increased expression of p-STAT3 in CRSwNP was observed relative to the control. Taken together, these data showed that the JAK2-associated cytokines including IL-6 and G-CSF may stimulate JAK2 phosphorylation to activate p-STAT3, indicating an association with disease severity and supporting its development of JAK2 inhibitor as a potential therapeutic agent for CRS.

## 1. Introduction

Chronic rhinosinusitis (CRS) is a common disease defined as chronic inflammation in the paranasal sinus mucosa [1]. CRS is subclassified as CRS with nasal polyps (CRSwNP) or without nasal polyps (CRSsNP) based on clinical and inflammatory patterns determined by nasal endoscopy or sinus computed tomography (CT) scan [2]. In the USA, the treatment cost is estimated at $22 billion USD per year for CRS [3], but therapeutic treatments are often ineffective, especially in patients with severe inflammatory endophenotypes [4]. Similarly, CRS has a significant impact on healthcare costs and patients’ quality of life in Taiwan [5]. The refractory CRSwNP has a poor response after topical or systemic corticosteroids treatment or combined surgery; therefore, the search for an endophenotypes-associated mechanism and development of clinically effective therapeutic interventions are essential work.

There is a wealth of evidence indicating that Janus kinase (JAKs)/activator of transcription proteins (STATs) signaling plays an important role in chronic lower respiratory tract inflammatory conditions, such as asthma and chronic obstructive pulmonary disease (COPD) [6]. Many cytokines are involved in chronic lower respiratory tract inflammatory diseases such as asthma, including interleukin (IL)-4, IL-5, IL-6, IL-9, IL-12, IL-13, granulocyte colony stimulating factor (G-CSF), granulocyte-macrophage colony stimulating factor (GM-CSF), and interferon (IFN)-γ, all of which activate JAK/STAT signaling [6,7,8]. The receptors for these cytokines activate JAKs by phosphorylation of STAT1, STAT3, STAT4, STAT5, and STAT6, suggesting that the JAK/STAT pathway is involved in the pathogenesis of chronic lower respiratory tract inflammatory diseases [7,9,10]. In a study on JAK/STAT signaling in CRS, Linke et al. revealed distinct expression and activation of STAT5b in nasal polyps, suggesting that STAT5b may contribute to the development of nasal polyps [11]. Moreover, dupilumab blocks both IL-4 and IL-13 signaling in treatment for patients with moderate to severe CRSwNP, thereby modulating downstream STAT6 or STAT3 phosphorylation via inactivation of JAK1 or JAK2; subsequently inflammation and tissue remodeling is able to be controlled and regulated [12]. Therefore, JAKs/STATs signaling could serve as a novel pathway in chronic upper airway inflammatory diseases, including treatment for CRSwNP. However, it remains unclear how JAKs/STATs signaling is related to CRSwNP disease severity. In addition, some reports have indicated that CRSwNP patients in Asia (specifically Chinese populations) show differences in cytokine patterns and inflammatory cells compared with Western patients [13,14,15].

In our previous study, we investigated the members of JAKs phosphorylation, such as p-JAK1, p-JAK2, and p-JAK3 by ELISA. Our preliminary data showed that the p-JAK2 level in patients’ NP tissue of CRSwNP was higher than in a control; however, there was no significant difference of p-JAK1 or p-JAK3 between each group with our collected subjects (control and CRSwNP). Therefore, in the present study, we investigated the JAK2-associated cytokine expression and JAK2-mediated STATs signaling phosphorylation in control and CRSwNP samples, as well as assessing their relationships with disease severity. The results may expand our understanding of JAK2-relevant signaling mediated by the JAK2-associated cytokines in CRSwNP of the Taiwanese population.

## 2. Materials and Methods

### 2.1. Subjects and Symptom Severity Assessment

This study was approved by the Institutional Review Board of the Chang Gung Medical Foundation (approval numbers 201701016B0 and 201900340B0). Eighty-seven patients who visited the Department of Otolaryngology of Kaohsiung Chang Gung Memorial Hospital in Taiwan between September 2017 and March 2020 were enrolled. We assessed nasal polyps from 61 CRSwNP patients who received endoscopic sinus surgery, and 13 inferior turbinate mucosa and 13 middle turbinate mucosa (a small piece of tissue by biopsy) from control subjects who underwent septomeatoplasty for relief of nasal obstruction with non-allergic chronic rhinitis, excluding previous sinonasal surgery, nasal tumor, other sinonasal disease. The diagnosis of CRS was based on the criteria outlined by the 2020 European position paper on rhinosinusitis [2], and CRS subjects were divided into two groups according to the Lund–Kennedy endoscopic grading system [16]. The parameters were as follows: presence of polyps, nasal mucosa edema, and presence of secretion. For each one of item scored 0 to 2, the sum of values obtained in both sides as the score ranged from 0–12. Therefore, the total score of 6–9 as mild/moderate CRSwNP [CRSwNP(I), *n* = 22] and total score of 10–12 as moderate/severe CRSwNP [CRSwNP(II), *n* = 39] respectively, were identified in our study. The following subjects were excluded: (1) patients with allergic fungal rhinosinusitis (AFRS) and aspirin-exacerbated respiratory disease (AERD); (2) patients who had taken systemic corticosteroids or immunomodulating drug therapy within 12 weeks before surgery; (3) patients who had underlying immunologic gastrointestinal, renal, endocrine, or skeletal disorders that might affect immune responses (such as multiple myeloma, rheumatoid arthritis, immunodeficiency, cystic fibrosis, and ciliary dyskinesia); and (4) patients with recurrent CRS. Clinical data were collected, including demographic information, patient history, smoking, and asthma status. For assessment of nasal polyp symptom severity, (1) the sinus computed tomography (CT) scores were based on the Lund–Mackay scale (LMK-CT) scores [17]. Each paranasal sinus was graded from 0 to 2 depending on the level of opacification. Total score was 0–24 points, and the highest value corresponded to greater severity of the disease. (2) Total nasal endoscopic polyp score (TPS) was used to grade the extent/severity of nasal polyps by nasal endoscopy. Each nostril was scored on a scale of 0 to 4, with the total score representing the sum of left and right nostril scores (range: 0–8) [18,19]. (3) Twenty-two-item sinonasal outcome test (SNOT-22) questionnaires were translated into Chinese for native Chinese speakers. Patients recalled their experience over a period of 2 weeks; their allergic symptoms and diagnoses were based on a severity scale from 0 to 5, with a total maximum score of 110 points [20]. 

### 2.2. Measurement of Cytokines

Cytokine measurement was performed simultaneously using the Bio-Plex Human Cytokine Group Assay kit (Bio-Rad Laboratories, Hercules, CA, USA) in accordance with the manufacturer’s instructions. In brief, 50 μL of antibody-coupled beads per well were added to a flat-bottom plate and washed twice. Then, the samples (50 μL) were incubated with antibody-coupled beads for 30 min at room temperature. After washing three times to remove unbound material, the beads were incubated with 25 μL of biotinylated detection antibodies for 30 min at room temperature. Three washes were carried out to remove unbound biotinylated antibodies, and the beads were then incubated with 50 μL of streptavidin-PE for 10 min at room temperature. Following removal of excess streptavidin-PE by three wash cycles, the beads were re-suspended in 125 μL of assay buffer. The beads were then read on the Bio-Plex suspension array system, and the data were analyzed using Bio-Plex Manager software version 6.0 (Bio-Rad Laboratories).

### 2.3. Enzyme-Linked Immunosorbent Assay

Human p-JAK2 (tyrosine-protein kinase JAK2) were assayed using ELISA kits (Fine Biological Technology, Wuhan, China). These kits were based on sandwich enzyme-linked immunosorbent assay (ELISA) technology. Anti-p-JAK2 antibodies were pre-coated onto 96-well plates. Biotin-conjugated anti-p-JAK2 antibodies were used as detection antibodies. Test samples (100 μL) and biotin-conjugated detection antibodies (100 μL) were added to the wells and incubated at 37 °C with a cover for 90 min. They were then washed with wash buffer three times. Horseradish peroxidase (HRP)-streptavidin conjugate (SABC) working solution was added into each well at a volume of 100 μL prior to incubation at 37 °C for 30 min. TMB substrates (90 μL) were kept in the dark at 37 °C for 15–30 min to visualize HRP enzymatic reactions. TMB catalyzed by HRP produced a blue-colored product that turned yellow following the addition of acidic stop solution (50 μL), with the density of the yellow color being proportional to the concentration of p-JAK2 in the sample. OD absorbance at 450 nm was measured using a microplate reader (BioTek Instruments, Inc., Winooski, VT, USA), and the concentrations of p-JAK2 were calculated as follows: (relative OD 450 nm) = (OD 450 nm in each well)—(OD 450 nm in the control well).

### 2.4. Tissue Microarray

Tissue microarrays were performed using the SIDSCO-TMA70 system (Scientific Integration Design Service Corp., Kaohsiung, Taiwan). Two tissue blocks were prepared for each case. In addition to hematoxylin and eosin staining, the sections were immunostained with the following primary antibodies: p-JAK2 (1:100, ab108596, Abcam, Cambridge, UK), p-STAT1 (1:200, #9167s, Cell Signaling Technology, Danvers, MA, USA), p-STAT3 (1:200, #9145s, Cell Signaling Technology), p-STAT5 (1:400, #9314s, Cell Signaling Technology), and p-STAT6 (1:100, ab28829, Abcam, Cambridge, UK). Sections were then incubated with goat anti-rabbit HRP. Finally, all slides were mounted with xylene-based mounting medium and scanned at 400× using the MoticEasyScan Pro (Motic in Asia, Xiamen, China). The quantitative intensity of staining was analyzed using Image-Pro Plus software (Version 6.0, Media Cybernetics, LP, USA). The immunohistochemistry intensity was determined as follows: staining percentage ≤25% positive cells as weak staining (-); 26–50% positive cells as mild to moderate staining (+/++); and >65% positive cells as intense staining (+++).

### 2.5. Statistical Analysis

Statistical analysis was performed using GraphPad Prism 5.0 software (GraphPad Software Inc., La Jolla, CA, USA). Data were presented as the means and standard error (mean ± SE). Differences among groups were tested by one-way analysis of variance with Bartlett’s test for equal variance and a Chi-square test, while a Pearson’s correlation test was used to determine correlations. In all cases, differences were considered statistically significant at *p* < 0.05.

## 3. Results

### 3.1. Patient Demographics

Tissue was enrolled from 61 CRSwNP patients and 26 control patients. Twenty-two mild NP as CRSwNP(I), 39 severe NP as CRSwNP(II), and 13 from inferior turbinate mucosa as control 1 and 13 from middle turbinate mucosa as control 2 were included in this study. Demographic information for each group, including the presence of atopy, asthma, diabetes, and smoking status are depicted in Table 1. There was no difference in gender, age, asthma, diabetes, or smoking status between all groups, but there were differences in atopy. There was a higher atopic IgE level among the two CRSwNP patients compared to the control 1 group (*p* < 0.001). Interestingly, there was no significant difference between control 2 and CRSwNP(I); in contrast, there was significant difference between control 2 and CRSwNP(II) group (Table 1).

### 3.2. p-JAK2 Level and JAK2-Associated Cytokines in CRS

The levels of p-JAK2 for the nasal polyp tissues were measured in each group. The results of ELISA determination, as shown in Figure 1A, demonstrate that the level of p-JAK2 in the CRSwNP(II) groups was significantly higher than in the two control groups (both *p* < 0.0001). There was a significantly higher p-JAK2 level in the CRSwNP(I) group compared with the control 1 (*p* = 0.0013) and the control 2 (*p* = 0.0302) groups, respectively. In addition, the level of p-JAK2 in the CRSwNP(II) groups was significantly higher than in the CRSwNP(I) (*p* = 0.0127). Similarly, the p-JAK2 expression in the epithelium of the polyps was evaluated by immunohistochemical staining. Data indicated that the p-JAK2 was significantly expressed in the epithelial basal layer of patients with CRSwNP(I); while p-JAK2 was highly expressed not only in both epithelial superficial and basal layer but also in inflammatory cells of patients with CRSwNP(II) (Figure 1B). However, we also evaluated the p-JAK1 and p-JAK3 by immunohistochemical staining, and no significant expression of p-JAK1 and p-JAK3 was found in each group in the present study (data not shown). Moreover, the levels of JAK2-associated cytokines for the nasal polyp tissues, including IL-2, IL-5, IL-6, IL-12, IL-13, G-CSF, GM-CSF, and IFN-γ, were measured by ELISA and presented in Table 2. In addition, cytokine comparisons between groups are shown in Table 3. The levels of T_H_ 2-associated cytokines, such as IL-5, IL-6, and IL-13 in the CRSwNP(I) and CRSwNP(II), were significantly higher than in the two control groups, and there were significant differences between the CRSwNP(I) and CRSwNP(II) groups. For the T_H_ 1-related cytokines, the amounts of IFN-γ in both CRSwNP groups were higher than in the control 1 group; and there were no significant differences among the CRSwNP(I) and CRSwNP(II) groups compared to the control 2 group; however, the values of IL-2 and IL-12 fell below the detectable range in all groups (data not shown). Regarding neutrophil-associated factors, such as G-CSF and GM-CSF, the level of G-CSF was significantly higher in both CRSwNP groups compared to the two control groups. However, there was no significant differences in each group for GM-CSF. Therefore, these findings showed that the levels of p-JAK2 and JAK2-associated cytokines were higher in CRSwNP patients than in the control groups, especially in severe NP such as CRSwNP(II) patients.

### 3.3. p-JAK2 and JAK2-Associated Cytokines Are Correlated with Disease Severity

The correlation between p-JAK2 or JAK2-associated cytokines and severity of disease is presented in Table 4. The Pearson correlation showed that a significantly positive relationship was found between the p-JAK2 level and all measurable scores for LMK-CT, TPS, and SNOT-22 (r = 0.45, 0.31, and 0.43, respectively). For JAK2-associated cytokines, there was strong significant correlation between the IL-5 or G-CSF level and all measurable scores for LMK-CT, TPS, and SNOT-22 (*p* < 0.05), while expression of IL-6 had a significant correlation with LMK-CT and SNOT-22 (r = 0.27 and 0.37, respectively). Moreover, there was a significant correlation between the IL-13 level and only one score for TPS score (r = 0.26). However, there was no correlation between the IFN-γ level and all measurable scores (Table 4).

### 3.4. JAK2/STATs Phosphorylation Signaling in Nasal Polyps

Immunohistochemical staining was performed to identify which the p-JAK2 downstream effectors of the p-STATs expression as presented in Figure 2. In the epithelium of the polyps for patients with CRSwNP(I) and CRSwNP(II), a significant expression of p-STAT3 was found in the nuclei of both epithelial cell and inflammatory cells. Staining intensity scores were calculated for p-STAT3 expression in each subject; these were 10 (45.45%) and 22 (56.4%) of intense intensity “+++”, respectively, in the CRSwNP(I) and CRSwNP(II) groups (*p* < 0.01 by Chi-square test). Moreover, the expressions of p-STAT5 and p-STAT6 were observed in inflammatory cells for the CRSwNP(II) group, with the staining intensity scores as mild/moderate intensity “+/++” were 16 (41.0%) and 7 (17.9%), respectively (Table 5). However, no significant difference was found in p-STAT1 expression between each group (data not shown). Taken together, positive signals for p-STAT3 were more frequent in the CRSwNP(I) and CRSwNP(II) groups. 

## 4. Discussion

Many studies have indicated that cytokines play a key role in chronic inflammation and structural changes of the respiratory tract by activating the JAK/STAT signaling cascade [6,7]. Differences in the profiles of inflammatory cytokines and immune cells have been identified in Chinese CRS patients compared with Caucasian CRS patients [15,21]. The present study investigated the JAK2/STATs pathway-associated cytokine pattern for Taiwanese CRSwNP patients. Nevertheless, there are some limitations in our study. We experienced difficulty in collecting the health nasal mucosa (IRB limitation), there was a low number of patients with CRSsNP in our clinical study, which may relate to medical habits of patients for Southern Taiwan. Our results found that the p-JAK2 have a high level in CRS, particularly in severe nasal polyps patients, discovered not only in the ELISA assay but also in IHC staining (Figure 1). Consequently, the JAK2-associated cytokines including IL-2, IL-5, IL-6, IL-12, IL-13, and IFN-γ were detected, and although IL-2 and IL-12 concentrations were non-detectable in all groups; IL-5, IL-6, IL-13, and IFN-γ levels were markedly higher in severe CRSwNP than in the control groups. The expression of p-JAK2 in nasal tissue had a significant correlation with common clinical assessments of disease severity (LMK-CT, TPS, and SNOT-22). Similarly, the T_H_2 cytokines such as IL-5, IL-6, and IL-13 levels highly correlated with the above-mentioned scores (Table 4). A wealth of evidence indicates that T_H_2 cytokines control the major components of the inflammatory response, including IgE isotype switching, mucus production, and the recruitment and activation of eosinophils [6,22], as well as regulating the phosphorylation of JAK2/STAT signaling [7]. In addition, IL-5 stimulates JAK2 to activate STAT5A/B [23] and IL-6 stimulates JAK1/JAK2 to activate STAT3 [24]. The conjugation of IL-13 with its receptor leads to the activation of JAK2 and TYK2 and the subsequent phosphorylation of STAT6 [25]. The T_H_1 cytokines such as IL-12 and IFN-γ are also involved in inflammatory disease by JAK/STAT signaling; which activates JAK2 and JAK1/JAK2 to phosphorylate the transcription factors STAT4 and STAT1 [26,27]. 

GM-CSF and G-CSF are both JAK2-associated cytokines, with GM-CSF triggering JAK2 to activate STAT5, which appears to govern differentiation and the inflammatory signature [6,28,29]. Moreover, G-CSF is an important biomarker for lung inflammation in patients with COPD or those predisposed to developing COPD [30]. A recent report indicated that G-CSF is known to activate the phosphorylation of STAT3 via JAK2 [31,32] and is involved in the maintenance of normal circulating granulocyte counts and in the neutrophilic response during infection [33]. Interestingly, our results showed that GM-CSF was detected at low concentrations. By contrast, the level of G-CSF in the CRSwNP(II) groups (mean = 2079.2 pg/mL) was more than 9 or 10 times higher than in the control 1 (mean = 187.3 pg/mL) and control 2 (mean = 267.4 pg/mL) groups. Similar results were reported by König et al. in 2016, who found that the G-CSF level (mean = 277 pg/mL) in nasal fluid of CRSwNP patients was three times higher than in the control (mean = 90 pg/mL) [34]. Furthermore, our data indicated that the G-CSF level was significantly correlated with disease severity (LMK-CT, TPS and SNOT-22) in CRSwNP patients. Based on immunohistochemistry, the expression of p-STAT3 was much more frequently observed in the CRSwNP(I) and CRSwNP(II) groups. A few CRSwNP patients had positive signals for p-STAT1, p-STAT5, and p-STAT6, and no statistically significant effects were identified (Figure 2 and Table 5). STATs as a key molecule of JAK signaling downstream have been investigated as a potential target in the treatment of inflammation disease and autoimmunity. A previous study indicated that IL-6 signaling may play a pathogenic role in CRSwNP [35]. In addition, p-STAT3 was found in both the superficial and basal layer of the epithelium of the polyps, which plays a crucial role in the proliferative development of nasal polyps [36]. Moreover, Lai et al. in 2019 indicated that IL-19 may upregulate mucin production via the STAT3 pathway in CRS [37]. Collectively, these results suggest that IL-6 or G-CSF activates the phosphorylation of JAK2/STAT3 and may be related to disease severity in CRSwNP patients.

CRS is a complex disease with the pathogenesis having both genetic and environmental components [38]. In Taiwan, there are special conditions, such as air pollution, a humid environment, and a particular ethnic genotype, which could explain differentiation from other inflammatory signatures of CRS around the world, with less eosinophilic and more neutrophilic inflammation found in Asian populations compared with Western populations [15]. Some studies have examined the cytokine patterns of CRS in the Taiwanese populations [39,40,41], but no previous reports have investigated JAK/STAT signaling in CRSwNP for the Taiwanese. Furthermore, many studies focus on development for JAK inhibition (Jakinibs) and novel mechanisms of the JAK/STAT signaling blockade [12,42], and successful use in autoimmune and inflammatory diseases [42].

In conclusion, our results suggested that the JAK2 associated-cytokines, including IL-5, IL-6, IL-13, G-CSF, and IFN-γ, displayed mixed type (T_H_1, T_H_2, and neutrophil-associated cytokines) immune response in severe CRSwNP patients. In addition, IL-6 and G-CSF may mediate the p-JAK2/p-STAT3 signaling which was correlated with disease severity. Further studies are needed to investigate the therapeutic potential of p-JAK2 inhibitors in CRSwNP through in in vitro or in vivo studies. Moreover, clarification is needed as to whether phosphorylation of JAK/STAT signaling involved refractory CRSwNP or not, such as recurrent CRSwNP.

## Figures and Tables

**Figure 1 biomolecules-11-01059-f001:**
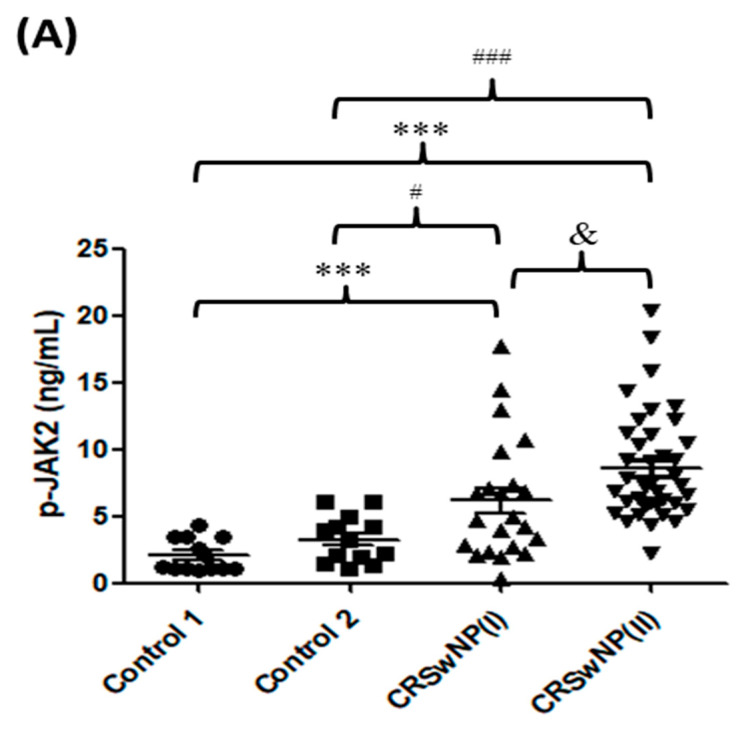
The p-JAK2 expression of the nasal polyp tissues in control 1 (*n* = 13), control 2 (*n* = 13), CRSwNP(I) (*n* = 22), and CRSwNP(II) (*n* = 39) groups was analyzed by (**A**) ELISA assay (**B**) immunohistochemistry staining. *** *p* < 0.01 vs. control 1; # *p* < 0.05 vs. control 2; ### *p* < 0.01 vs. control 2; and & *p* < 0.05 vs. CRSwNP(I) group. Black arrows: epithelial cell staining; and red arrows: inflammatory cells staining. Scale bar = 50 μm.

**Figure 2 biomolecules-11-01059-f002:**
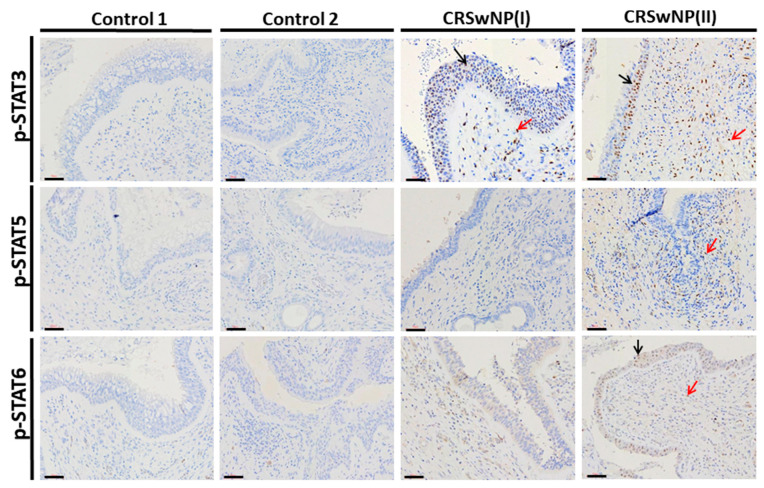
The expression of p-STAT3, p-STAT5, and p-STAT6 for the nasal polyp tissues in control 1, control 2, CRSwNP(I), and CRSwNP(II) groups evaluated by immunohistochemistry. Black arrows: epithelial cell staining; and red arrows: inflammatory cells staining. Scale bar = 50 μm.

**Table 1 biomolecules-11-01059-t001:** Demographics of patients with control groups and CRSwNP groups.

	Subjects				
Demographics	Control 1 (*n* = 13)	Control 2 (*n* = 13)	CRSwNP(I) (*n* = 22)	CRSwNP(II) (*n* = 39)	*p*-Value
Gender (female), *n* (%)	5 (38.5)	7 (53.8)	8 (36.4)	8 (20.5)	0.131
Age (years), mean ± SE	40.2 ± 3.6	36.6 ± 3.2	45.4 ± 2.6	49.4 ± 2.6	0.398
Asthma, *n* (%)	0 (0)	0 (0)	3 (13.6)	5 (12.8)	0.288
Smoking, *n* (%)	2 (15.3)	3 (23.1)	4 (18.2)	6 (15.4)	0.930
Diabetes, *n* (%)	1 (7.7)	1 (7.7)	2 (9.1)	3 (7.7)	0.998
Peripheral eosinophil (%), mean ± SE	3.2 ± 0.8	3.6 ± 0.7	4.1 ± 0.5	3.3 ± 0.4	0.774
Atopic IgE (KU/L), mean ± SE	65.4 ± 18.1	76.9 ± 19.3	115.2 ± 17.1	179.8 ± 36.4	<0.001

CRS: chronic rhinosinusitis; NP: nasal polyps. Control 1 from inferior turbinate mucosa. Control 2 from middle turbinate mucosa. *p* < 0.05 were considered statistically significant differences.

**Table 2 biomolecules-11-01059-t002:** JAK2-associated cytokines of nasal polyp tissues in control groups and CRSwNP groups.

Cytokines	Control 1 (C1)	Control 2 (C2)	CRSwNP(I)	CRSwNP(II)	*p*-Value
Number	13	13	22	39	
Mean ± SE (Range)					
IL-5	12.23 ± 1.40 (4.88–21.26)	13.24 ± 1.56 (4.13–25.87)	36.84 ± 4.82 (10.05–90.86)	56.86 ± 4.32 (10.96–112.34)	<0.001
IL-6	51.77 ± 10.89 (23.07–164.01)	129.77 ± 46.51 (22.56–518.56)	255.34 ± 63.53 (24.51–976.91)	356.81 ± 43.97 (142.40–1073.57)	0.009
IL-13	7.74 ± 1.79 (1.51–22.77)	12.67 ± 1.53 (2.77–24.9)	25.79 ± 3.92 (5.28–82.27)	40.10 ± 2.86 (12.91–95.74)	<0.001
G-CSF	187.31 ± 47.28 (45.33–558.07)	267.39 ± 98.87 (54.79–1032.31)	1045.71 ± 203.12 (69.83–3579.17)	2079.17 ± 331.99 (71.06–8847.96)	<0.0001
GM-CSF	0.90 ± 0.24 (0.09–2.3)	1.50 ± 0.19 (0.13–2.73)	1.51 ± 0.21 (0.11–2.78)	1.6 ± 0.20 (0.12–3.43)	0.2654
IFN-γ	34.83 ± 9.37 (5.78–113.43)	92.43 ± 11.33 (15.28–172.53)	98.5 ± 15.90 (16.12–330.92)	119.59 ± 10.08 (14.15–363.48)	0.006

Concentrations are given in pg/mL. CRS: chronic rhinosinusitis; NP: nasal polyps; IL: interleukin; G-CSF: granulocyte colony-stimulating factor; GM-CSF: granulocyte-macrophage colony stimulating factor; IFN-γ: interferon-gamma. Control 1 from inferior turbinate mucosa. Control 2 from middle turbinate mucosa. *p* < 0.05 were considered statistically significant differences.

**Table 3 biomolecules-11-01059-t003:** Comparison of *p*-value between control groups and CRSwNP groups.

	Cytokines					
Groups	IL-5	IL-6	IL-13	G-CSF	GM-CSF	IFN-γ
Control 1 vs. Control 2	0.6816	0.0812	0.0455	0.7976	0.1177	0.0012
Control 1 vs. CRSwNP(I)	<0.001	0.0039	0.003	0.0016	0.1419	0.0025
Control 1 vs. CRSwNP(II)	<0.001	<0.001	<0.001	<0.0001	0.0869	<0.001
Control 2 vs. CRSwNP(I)	0.01	0.0187	0.0212	0.006	0.8643	0.9184
Control 2 vs. CRSwNP(II)	<0.001	0.0008	<0.001	<0.0001	0.8326	0.1663
CRSwNP(I) vs. CRSwNP(II)	0.0046	0.0124	0.008	0.0396	0.7753	0.1219

CRS: chronic rhinosinusitis; NP: nasal polyps; IL: interleukin; G-CSF: granulocyte colony-stimulating factor; GM-CSF: granulocyte-macrophage colony stimulating factor; IFN-γ: interferon-gamma; TNF-a: tumor necrosis factor alpha. Control 1 from inferior turbinate mucosa. Control 2 from middle turbinate mucosa. *p* < 0.05 were considered statistically significant differences.

**Table 4 biomolecules-11-01059-t004:** Linear regression analyses with Pearson correlation coefficients comparing p-JAK2 or its associated cytokines and clinical measures of disease severity in CRSwNP.

	Clinical Assessments of Disease Severity								
	LMK-CT			TPS			SNOT-22		
Expression	r	*p*-Value	95% CI	r	*p*-Value	95% CI	r	*p*-Value	95% CI
p-JAK2	0.45	0.0002	0.192–0.598	0.31	0.0139	0.112–0.955	0.43	0.0005	0.0458–0.155
IL-5	0.42	0.0007	1.050–3.677	0.30	0.0259	0.456–6.86	0.46	0.0002	0.0586–0.763
IL-6	0.27	0.0344	1.204–30.41	0.09	0.4764	−52.70–111.50	0.37	0.038	1.900–9.414
IL-13	0.23	0.0699	−0.0772–1.92	0.26	0.041	0.0870–4.05	0.23	0.0751	−0.0251–0.506
G-CSF	0.44	0.0003	78.01–251.1	0.34	0.0073	68.72–423.3	0.38	0.0028	13.35 to 60.94
IFN-γ	0.13	0.3259	−1.80–5.34	0.04	0.7351	−6.146–8.65	0.05	0.7118	−0.779–1.133

CRS: chronic rhinosinusitis; NP: nasal polyps; IL: interleukin; G-CSF: granulocyte colony-stimulating factor; IFN-γ: interferon-gamma; TNF-a: tumor necrosis factor alpha; LMK-CT: Lund–Mackay computed tomography; TPS: total nasal endoscopic polyp score; 95% CI: 95% confidence intervals. *p* < 0.05 were considered statistically significant differences.

**Table 5 biomolecules-11-01059-t005:** Distribution by immunohistochemistry of intensity for p-STAT1, p-STAT3, p-STAT5, and p-STAT6 in control groups and CRSwNP groups.

				Staining Intensity, *n* (%)								
	p-STAT3 **			p-STAT5			p-STAT6			p-STAT1		
Subjects	-	+/++	+++	-	+/++	+++	-	+/++	+++	-	+/++	+++
Control 1 (*n* = 13)	92.3 (12)	7.7 (1)	0.0 (0)	100.0 (13)	0.0 (0)	0.0 (0)	100.0 (13)	0.0 (0)	0.0 (0)	100.0 (13)	0.0 (0)	0.0 (0)
Control 2 (*n* = 13)	84.6 (11)	15.4 (2)	0.0 (0)	100.0 (13)	0.0 (0)	0.0 (0)	100.0 (13)	0.0 (0)	0.0 (0)	100.0 (13)	0.0 (0)	0.0 (0)
CRSwNP(I) (*n* = 22)	9.1 (2)	45.45 (10)	45.45 (10)	81.8 (18)	18.2 (4)	0.0 (0)	90.9 (20)	9.1 (2)	0.0 (0)	100.0 (22)	0.0 (0)	0.0 (0)
CRSwNP(II) (*n* = 39)	5.1 (2)	38.5 (15)	56.4 (22)	59.0 (23)	41.0 (16)	0.0 (0)	79.5 (31)	17.9 (7)	2.6 (1)	97.4 (38)	2.6 (1)	0.0 (0)

CRS: chronic rhinosinusitis; NP: nasal polyps. “-”: weak; “+/++”: intensity mild/moderate; “+++”: intensity intense. Control 1 from inferior turbinate mucosa. Control 2 from middle turbinate mucosa. ** *p* < 0.01 were considered statistically significant by Chi-square test.

## Data Availability

The data presented in this study are available on request from the corresponding author.

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
