# Peer review of "JAK2 Phosphorylation Signals and Their Associated Cytokines Involved in Chronic Rhinosinusitis with Nasal Polyps and Correlated with Disease Severity"

_biomolecules, 2021, doi:10.3390/biom11071059_

Round 1
Reviewer 1 Report
The authors reported JAK2 phosphorylation signals and its associated cytokines may involve in chronic rhinosinusitis with nasal polyps and correlated with disease severity. The logic of this paper is quite straight forward, the authors have demonstrated JAKs phosphorylation signals are correlated to clinical samples of chronic rhinosinusitis ,
Some comments:
In the last paragraph of the Introduction which mentioned the preliminary findings on p-JAK2, is there any published reference(s)?
For the microtissue scoring method for the phospharylated proteins, it is only semi-quantitative and seems to be subjective, is it possible to use some more objective calculation method such as calculating the intensity of the signals using software.
Reviewer 2 Report
The authors investigated jak expression in a representative cohort of CRSwNP patients. This is an interesting topic considering jay inhibition is already a known therapeutic target other type 2 inflammatory diseases.
Please clarify
- Why endoscopic grading according to Lund Mackay score (but not complete LM score) was chosen to group patients CRSwNP severity. If only polyp size was considered as marker of severe disease TPS would be appropriate
- if all patients had primary surgery for CRSwNP. If patients treated for recurrent disease nearly recurrence might be a more predictive marker for severe disease than polyp size
- why markers of type 2 inflammation (tissue Eos, blood Eos, IgE where not correlated to jak expression
- why expression was not compared to CRSsNP patients, authors seem to address this problem in the discussion but explanation is not sufficient
Please cite EPOS 2020 (instead of 2012) and adapt accordingly
